# Synthesis and Human Carbonic Anhydrase I, II, IX, and XII Inhibition Studies of Sulphonamides Incorporating Mono-, Bi- and Tricyclic Imide Moieties

**DOI:** 10.3390/ph14070693

**Published:** 2021-07-19

**Authors:** Kalyan K. Sethi, KM Abha Mishra, Saurabh M. Verma, Daniela Vullo, Fabrizio Carta, Claudiu T. Supuran

**Affiliations:** 1Department of Medicinal Chemistry, National Institute of Pharmaceutical Education and Research Guwahati, Assam 781101, India; mishraabha590@gmail.com; 2Department of Pharmaceutical Sciences and Technology, Birla Institute of Technology, Mesra, Ranchi 835215, India; smverma@bitmesra.ac.in; 3Neurofarba Department, Università degli Studi di Firenze, Sezione di Farmaceutica e Nutraceutica, Via Ugo Schiff 6, Sesto Fiorentino, 50019 Florence, Italy; daniela.vullo@unifi.it (D.V.); farizio.carta@unifi.it (F.C.)

**Keywords:** human carbonic anhydrase inhibitors, benzenesulphonamide, anhydride, docking, SAR

## Abstract

New derivatives were synthesised by reaction of amino-containing aromatic sulphonamides with mono-, bi-, and tricyclic anhydrides. These sulphonamides were investigated as human carbonic anhydrases (hCAs, EC 4.2.1.1) I, II, IX, and XII inhibitors. hCA I was inhibited with inhibition constants (Kis) ranging from 49 to >10,000 nM. The physiologically dominant hCA II was significantly inhibited by most of the sulphonamide with the Kis ranging between 2.4 and 4515 nM. hCA IX and hCA XII were inhibited by these sulphonamides in the range of 9.7 to 7766 nM and 14 to 316 nM, respectively. The structure–activity relationships (SAR) are rationalised with the help of molecular docking studies.

## 1. Introduction

The zinc metalloenzymes carbonic anhydrases (CAs, EC 4.2.1.1) are ubiquitously found in most of the living creatures. CAs are encoded by eight genetically distinct CA families [1,2,3]. Human CAs (hCAs) belong to the α-CA. There are 15 such isoforms, which differ by the oligomeric arrangement, three dimensional structures, location in cell, tissues distribution, and catalytic properties [2]. There are five cytosolic, five membrane bound, two mitochondrial, and a secreted hCA isozyme [1,2,3]. Three acatalytic hCA isoforms are present whose functions are less well understood [3]. hCAs catalyses the interconversion between CO_2_ and HCO_3_, being involved in many physiological processes, including breathing, transport of CO_2_/HCO_3_^−^, pH homeostasis, biosynthetic reactions, electrolyte secretion, and calcification. hCA inhibitors usually bind to the Zn^2+^ ion, although compounds possessing different inhibition mechanisms were also described [1,2,3].

Several studies demonstrated that abnormal expression levels of hCA enzymes are associated with different human diseases, making them a prominent target for the design of modulators of their activity with biomedical applications [2,4,5,6,7]. hCA inhibitors were clinically used as diuretics, for the management of glaucoma, epilepsy, and altitude sickness, whereas their use in obesity, in the management of tumours and other pathologies were validated recently (Table 1) [8,9,10,11,12,13,14,15,16,17,18,19]. The larger number of hCA isoforms and the new applications of their inhibitors lead to the search for compounds with improved specificity and selectivity profile for the targeted isoform, to avoid severe adverse effects due to inhibition of other isoenzymes which are not relevant in the considered pathology [1,2,3]. The various hCA isoenzymes and their organ and tissue distribution, kinetics, and affinity towards primary sulphonamides are presented in Table 1. The 3D structure elucidation of many isoforms among which hCA I, II, IX, and XII has been reported [20] and showed a rather close similarity in structure, i.e., a central twisted *β*-sheet bounded by helical connections and added *β*-strands (Figure 1) [21].

The binding pocket is present in a large, conical cavity, which is ~12 Å wide and ~13 Å deep. The catalytic Zn^2+^ is present at the bottom of the cavity, being in a tetrahedral geometry, coordinated by three histidine residues and a water molecule/hydroxide ion as ligands (Figure 1) [21,22].

**Figure 1 pharmaceuticals-14-00693-f001:**
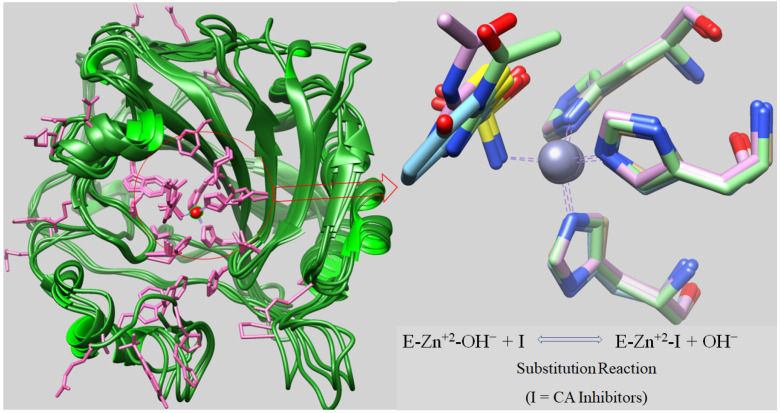
3D multiple superpositions of four α-hCAs crystal structures (high sequence alignment homology) [23]. The multiple superpositions involved the following crystal structures: 1AZM (hCA I); 1ZFQ (CA II); 3IAI (CA IX); and 1JD0 (CA XII) with the Zn ion shown as red sphere, and its coordinating residues His 94, His 96, and His 119 shown in pink. The protein backbone is shown in green [1,20].

hCAs I and II are involved in glaucoma, cerebral and macular oedema, epilepsy, and mountain sickness [8,9,10,17,18]. hCA IX and XII are validated targets for hypoxia-induced cancers, being strongly overexpressed in many types of such tumours [2,16].

CA inhibitors (CAIs) coordinating to the zinc ion have been developed by incorporating various zinc bindings groups [1]. The classical sulphonamide one, is still a key player in this field for the development of many classes of therapeutic agents [1,2]. Sulphonamides bind the Zn^2+^ ion of the enzyme by replacing the OH^−^ ion coordinated to the zinc (Figure 1) [20].

We report here the design of novel sulphonamide CAIs based on the “tail” approach [21] which consists of inserting moieties/fragments at the terminal part of the CAI molecule, to induce the desired selectivity or physico-chemical properties (Figure 2). Various drug design methods, i.e., ligand-based drug design (CoMFA/CoMSIA), pharmacophore-based drug design, and structure-based drug design (docking study) have been employed to design CAIs. It is rational drug design which helps to find new druggable targets, i.e., organic small molecule sulphonamides for enzyme inhibition. A validated CAI showing acquiescence to the overall pharmacophore feature of is shown in Figure 2 [21,22,23]. These approaches may produce more potent, more specific, acceptable water/lipid-soluble, and good penetrable hCA inhibitors. Thus, the tails selected to be incorporated for the synthesis of sulphonamides considered here are various aliphatic/aromatic anhydrides, which may lead to new pharmacological properties [22,23,24,25,26,27,28].

## 2. Results and Discussion

### 2.1. Chemistry

Several sulphonamide derivatives incorporating phthalimido moieties were reported earlier by our groups [21,22,23]. The compounds obtained from 3 and 4-nitrophthalic anhydride, 4,5,6,7-tetrabromophthalic anhydride, and 4,5,6,7-tetrachlorophthalic anhydride showed good activity, acting as nanomolar inhibitors against hCA I, II, IX, and XII [22,23]. Sulphonamides containing phthalimide moieties were reported to possess antiglaucoma activity in a rabbit model of the disease [22].

The synthesis of a series of 13 novel sulphonamides **1–13** is shown in Scheme 1. The anhydrides reacted with the free NH_2_ group of aromatic sulphonamides and formed the corresponding imides (Figure 3). The reaction between amino group and anhydrides involves S_N_^2^ type reaction mechanism which resulted in rearrangement via a transition state (Figure 3).

### 2.2. Carbonic Anhydrases Inhibition Studies

The hCA I, II, IX, and XII inhibition results of sulphonamides 1–13, are showed in Table 2. The inhibition studies of standard AAZ and other clinically used sulphonamides/sulfamate are also considered for comparison and development of SAR. AAZ is a clinically used drug for the adjunctive treatment of many diseases (Table 1) [8,9,10,17,18].

The following SAR was observed for this series of synthesised sulphonamides, after comparison with the clinically used drugs shown in Table 2:i.The synthesised sulphonamides inhibited the cytosolic hCA I, with inhibition constants (Ki) ranging from 49 nM to >10,000 nM (Table 2). The 4-(4-sulfo-1,8-napthalic-1,3-dioxopyridine) potassium benzene sulphonamide **1** (Ki of 49 nM) was the most potent inhibitor of the series, whereas the compound 3-chloro-4-(4,5,6,7-tetrachloro-1,3-dioxoisoindolin-2-yl) benzene sulphonamide 12 with an inhibition constant of 159 nM better than AAZ and TPM (Table 2). In comparison to BRZ, CLX, DZR, SLP, and VLX, all the synthesised compounds showed significant inhibition constant (Table 2). The compound **10** is the weakest inhibitor that can be contemplated as a positive feature of hCA I because it is abundantly found in RBCs and is undoubtedly an off-target for other CAIs [2,28]. Increasing the chain length of carbon between the benzene and bulky aromatic group (-CH_2_CH_2_- or -CH_2_-) leads to a decreased activity of compounds **7** and **8** with 427 nM to 332 nM, respectively compared to compound **6**. The position of -SO_2_NH_2_ plays an important role in increase or decrease of the activities. The decreased activity of compounds **6** > **9** > **10** according to position of -SO_2_NH_2_ in the benzene ring are 4th > 3rd > 2nd (Kis 332 nM to >10,000 nM). It is very clear that –SO_2_NH_2_ present in the 2^nd^ position comparatively very less active than 3rd than 4th due to steric hindrance. The electronegativity of halogen (F > Cl > Br) to the benzene ring was found to influence the hCA I inhibition activity of compounds **11** (368 nM), 12 (159 nM), and 13 (281 nM).ii.hCA II was significantly inhibited by many of the new sulphonamides, which exhibited Kis ranging from 2.4 to 4515 nM (Table 2). Most of the compounds showed significant inhibition constants than clinically used standard AAZ and other sulphonamide drugs. Compound **13** significantly inhibits hCA II with a Ki of 2.4 nM than the other sulphonamides of the series. Although the compounds showed excellent to moderate inhibition activity for hCA II. The SAR is straightforward. The increased chain length of carbon between the benzene and bulky aromatic group (-CH_2_CH_2_- or -CH_2_-) results in increased activities of the compounds **1** < **2** < **3** with Kis of 7.1 nM, 5.2 nM, and 2.9 nM, respectively. Due to steric hindrance, the hCA II inhibition activity was lost, i.e., the Ki of ortho, meta, and para position of –SO_2_NH_2_ in compound **10** < **9** < **6** was 4515, 27.7, and 7.1 nM, respectively). The electronegativity of halogen (F > Cl > Br) attached to the benzene ring was found to influence the hCA II inhibition of compounds **11** (3.4 nM), 12 (4.9 nM), and 13 (2.4 nM). Similarly, benzthiazole sulphonamide analogues **3** (742 nM) and 4 (44 nM). More electronegativity of halogen lesser is the potency/efficacy. Overall, most of these sulphonamides showed a potent action of inhibition against hCA II. hCA II is the main off-target isoform among several hCAs [28].iii.hCA IX was moderately to poorly inhibited by the sulphonamide (1–13) with Kis ranging from 9.7 to 7766 nM (Table 2). Compound **6** potentially inhibited hCA IX with a Ki of 9.7 nM than the standard AAZ. Although the sulphonamides are moderate to poorly effective against hCA IX, but the SARs are generated for the future development of more potent hCA IX inhibitors. The increase in the chain length of the carbon between the two bulky groups (-CH_2_CH_2_- or -CH_2_-) results in decreased activities of the compounds, i.e., compound **6** (Ki = 9.7 nM) is more potent than compounds **7** (Ki = 103 nM) and **8** (Ki = 53 nM). The steric hindrance effects cause a significant loss of the hCA IX inhibition properties of sulphonamides, i.e., –SO_2_NH_2_ at ortho, meta, and para in compound **9** (Ki ≥ 498 nM) < **10** (Ki = 559 nM) < **6** (Ki = 9.7 nM). The electronegativity of halogen (F > Cl > Br) in the benzene ring was found to be influencing the hCA IX inhibition of compounds **11** (88 nM), **12** (49 nM), and **13** (40 nM). More electronegative halogen atom attached to the benzene ring, reduce the hCA IX inhibition properties.iv.hCA XII was moderately inhibited by these sulphonamides (Table 2). The inhibition constant showed Kis ranging from 14 to 316 nM. Compound **11** was the most potent hCA XII inhibitor (Ki = 14 nM) of the series. The SARs of these sulphonamides were developed for hCA XII inhibition. The increased carbon chain length (-CH_2_CH_2_- or -CH_2_-) between benzene and anhydride decreased the activities of compounds **6** > **8** > **7** with Kis of 33 nM, 226 nM, and 300 nM, respectively. Steric hindrance responsible for the loss of hCA XII inhibition properties, i.e., –SO_2_NH_2_ present in the ortho, para, and meta substituted sulphonamides **9** < **10** < **6** (Kis in the range of 33, 309, and 316 nM), respectively. The electronegativity of halogen (F > Cl > Br) in the benzene ring was found to be influencing the hCA XII inhibition of compounds **11** (14 nM), **12** (22 nM), and **13** (99 nM). More the electronegativity of halogen attached to the benzene ring, increase the hCA XII inhibition properties.v.The close structural similarities in hCAs were the major challenge in finding novel selective hCA isoenzyme inhibitors. Thus, calculation of the selectivity ratios has been done (Table 2). The selectivity ratios for the tumour-associated isoforms hCA IX and XII over hCA II, ranged from 1 to 8.077 nM and 0.013 to 14.616 nM, respectively for the synthesised sulphonamide reported here (Table 2). Some compounds were observed to have high selectivity ratios for hCA IX and XII over hCA II. Compound **10** was highly selective for hCA IX inhibition over hCA II. Similarly, compounds **3** and **10** were highly selective for hCA XII inhibition over hCA II. However, most of the compounds had a low selectivity ratio, indicating that these are more selective for inhibition of hCA I and II than hCA IX and CA XII.

### 2.3. Docking Studies

All the synthesised sulphonamides and clinically used sulfamate/sulphonamides were docked to different hCA isoforms of interest, i.e., 1AZM (hCA I), 1ZFQ (hCA II), 3IAI (hCA IX), and 1JD0 (hCA XII) [20] to rationalize the observed SARs and to find out the binding mode and other interactions. The Extra Precision docking results of the co-crystallised ligand AAZ were between −3.8 to −4.8 (Table 3) and the RMSD values obtained ranging from 1.8–2.3 which is better for docking studies.

The docking results of the synthesised sulphonamides are listed in Table 3. The synthesised compounds showed good binding interactions in the catalytic site of all hCAs. For example, AAZ, the co-crystallised ligand showed a docking score of −4.4 with the hCA IX protein (pdb code: 3IAI) binding interactions with various amino acids. It formed a hydrogen bond with THR 199, THR 200, and two water molecules in the catalytic domain (Figure 4a). The NH^−^ of SO_2_NH_2_ interacted with Zn^2+^ (Figure 4a). Similarly, compound **1** showed the docking score of −7.749 with the hCA IX protein (pdb code: 3IAI) in which the H of -SO_2_NH_2_ bind to SER 6 (2.130 Å), O of C=O bind to HIS 64 (1.87 Å), and O of -C=O bind to THR 199 (2.156 Å). In most of the cases, it was observed that the SARs tally the docking results obtained. For example, compounds **8**, **6**, and **7** showed the hCA inhibition of 326 nM, 332 nM, and 427 nM, respectively. Comparing them to the docking scores, they are in the same order, i.e., compounds **8**, **6**, and **7** have the docking scores of −6.118, −5.827, and −5.314, respectively (Table 3). The selected compounds showed good binding interaction with the hCA I, II, IX, and XII.

Most abundant cytosolic hCA I, which is mainly present in the erythrocytes, GI tract, and the eye is strongly inhibited by compounds **1**, **3**, and **12** compared to the standard sulphonamide AAZ (Table 2). Thus, these compounds can be considered as candidates for the treatment of retinal/cerebral oedema, ischemic diabetic cardiomyopathy, coronary revascularization, diabetic retinopathy, etc. [8,9,18,31]. Similarly, hCA II, the cytosolic isozyme abundantly present in the erythrocytes, GI tract, eye, bone-osteoclasts, kidney, lung, testis, and brain [2] was significantly inhibited by compounds **1**, **5**, **6**, **7**, **8**, **11**, **12**, and **13** compared to the standard AAZ (Table 2). These compounds can be much more useful for the treatment of oedema [8], glaucoma [9], epilepsy [10], cancer [16], altitude sickness [17], and acute altitude sickness (AAS) [32].

Compound **6** also showed significant hCA IX inhibition activity. It can be further developed to a clinically meaningful agent for the treatment of hypoxia-induced cancer, i.e., gliomas/ependymomas, mesotheliomas, carcinomas of the bladder, uterine cervix, papillary/follicular carcinomas, nasopharyngeal carcinoma, head and neck, oesophagus, lungs, breast, brain, vulva, squamous/basal cell carcinomas, and kidney tumours [2,16,22]. The compounds are moderately potent for the inhibition of transmembrane hCA XII. The information would be very much useful for the design and development of more potent druggable molecules.

BRZ and DZA are potent hCA II inhibitors used clinically for the treatment of glaucoma or ocular hypertension. Compared to BRZ compounds **8** and **13** are more potent in terms of hCA II inhibition. Similarly, in comparison to DZA compounds **5**, **6**, **7**, **8**, and **11** are significantly inhibited hCA II (Table 2). Thus, these compounds can be developed as drug molecules for the treatment of glaucoma or ocular hypertension. TPM and ZNS are clinically used for the treatment of epilepsy, migraines, Parkinson’s disease, etc. Some of the sulphonamides are better than these clinically used hCA inhibitors (Table 2). Further pre-clinical and clinical trials are required to test these potent hCA inhibitors for their clinical importance [2,33,34,35,36]. Patients affected by COVID-19 could show a dysregulated acid–base status probably influenced by the CA activity, which is highly increased in patients affected by COVID-19 infection. Thus, these compounds could be useful as an adjunctive pharmacological treatment for COVID-19 [37].

## 3. Materials and Methods

### 3.1. Reagents and Instruments

All the reagents and solvents employed in this study were purchased in India and Italy. Para aminobenzene sulphanilamide (Sigma-Aldrich, Bangalore, India), 4-sulfo-1,8-napthalic anhydride potassium salt (Sigma-Aldrich, Bangalore, India), tetrachlorophthalic anhydride (Sigma-Aldrich, Bangalore, India), tetrabromophthalic anhydride (Sigma-Aldrich, Bangalore, India), maleic anhydride (Sigma-Aldrich, Bangalore, India), 4-nitrophthalic anhydride (Sigma-Aldrich, Bangalore, India), glacial acetic acid (Merck, Bangalore, India), chloroform, methanol (Merck, Bangalore, India), DMSO (CDH), and silica gel 60 F254 plates (Merck Art.1.05554, Bangalore, India) were procured in India. 2-amino benzene sulphanilamide, 3-amino benzene sulphanilamide, para methyl amino benzene sulphanilamide, para ethyl amino benzene sulphanilamide, 3-fluro para-amino benzene sulphanilamide, 3-chloro para-amino benzene sulphanilamide, and 3-bromo para-amino benzene sulphanilamide was arranged from Florence laboratory, Italy. TLC spots were observed under the wavelength of 254 nm and 365 nm and/or by using ninhydrin stain solution if required. FT-IR spectra were taken from Bruker, Alpha II and 8400S, Shimadzu. ^1^H and ^13^C NMR spectra were characterised with the help of Bruker DRX-400 spectrometer, where DMSO-d6 was used as solvent and tetramethylsilane for the internal standard. Chemical shifts of ^1^H NMR spectra are expressed in δ (ppm) downfield from tetramethylsilane and coupling constants (J) are expressed in Hertz. High-resolution electron ionization mass spectra were performed with the help of Water MicroMass ZQ, Waters Alliance, Newtown, PA, USA. CHNSO elemental analysis was performed with the help of Elementar, Vario EL III (Thermo Scientific™, Santa Clara, CA, USA).

### 3.2. Chemistry

Benzothiazole sulphonamide (6-sulfamido-2-amino-1,3-benzothiazole) was synthesised using benzene sulphonamide and potassium thiocyanate in brominated glacial acetic acid. The mixture was stirred for 20 h to get yellow coloured benzothiazole sulphonamide product. The reaction mixture was then added into cold water, made alkaline with 50% aqueous ammonia, and precipitated solids were filtered out and dried. The dried solids obtained were re-crystallised with dry ethanol [38]. After that, desired sulphonamide and respective anhydride in glacial acetic acid were refluxed and stirred under the nitrogen environment for a suitable time. The reactions were monitored from time to time with the help of TLC till the completion of the reaction. A total of compounds **1–13** were synthesised (Scheme 1) [22,23,24,25,26].

#### 3.2.1. Synthesis of 4-(4-Sulfo-1,8-napthalic-1,3-dioxopyridine) Potassium Benzenesulphonamide (**1**)

A reaction containing 0.002 moles (0.344 g) of 4-aminobenzenesulphonamide and 0.002 moles (0.631 g) of 4-sulfo-1,8-napthalic anhydride potassium salt in glacial acetic acid (as solvent) was stirred at 130 °C for 12 h under the nitrogen environment to produce 0.002 moles of 4-(4-sulfo-1,8-napthalic-1,3-dioxopyridine) potassium benzenesulphonamide (**1**) (0.939 g). The reaction was checked every 30 min by using TLC (solvent system—chloroform: methanol; 1:1). After 12 h and the confirmation of the completion of the reaction by TLC, the reaction mixture was then cooled, and cold water (30 mL) was added. Filtration of the product followed by washing with cold water frequently 4 to 5 times and further recrystallization in ethanol provided compound [22,23,24,25,26].

White crystalline and solid; yield% = 80%; m.p. = 277 °C; solubility; insoluble: acetic acid glacial; partially soluble: ethanol; fully soluble: water, DMSO and methanol, chloroform. IRv_max_ (cm^−1^; KBr pellets); 1783.25, 1770.71 (C=O imide); 1304.89, 1159.26 (S=O) and 3525.99 (NH_2_), 9.40. Mass (ESI+); m/z: 431.0812 [M-K]^+^. ^1^H NMR (400 MHz, DMSO−d_6_): δ 7.441 (s, 2H, SO_2_NH_2_), 7.848−7.888, 7.91−7.931 (d, 2H, Ar−H from benzenesulphonamide), 7.580−7.601, 8.192–8.211, 9.264–9.286 (d, 3H, Ar−H from napthalic ring), 8.416−8.462 (t, 2H, Ar−H from napthalic ring).^13^C NMR (100 MHz, DMSO−d_6_): 123.36, 124.03, 125.95, 127.30, 127.71, 128.66, 129.53, 130.88, 131.17, 131.38, 135.38, 140.03, 144.80, 151.11, 164.24, 164.68. Elem. Anal. calculated for C_18_H_11_KN_2_O_7_S_2_: C, 45.95; H, 2.36; K, 8.31; N, 5.95; O, 23.80; S, 13.63; found C, 45.91; H, 2.35; N, 5.91; O, 23.74; S, 13.62.

#### 3.2.2. Synthesis of 4-(2-(2,5-Dioxo-2H-pyrrol-1(5H)-yl) Ethyl) Benzenesulphonamide (**2**)

A reaction containing 0.002 moles (0.401 g) of 4-(2-aminoethyl) benzenesulphonamide and 0.002 moles (0.196 g) of furan-2,5-dione (maleic anhydride) in glacial acetic acid (as solvent) was stirred at 130 °C for 12 h under the nitrogen environment to produce 0.002 moles of 4-(2-(2,5-dioxo-2H-pyrrol-1(5H)-yl) ethyl)benzenesulphonamide (**2**) (0.560 g). The reaction was checked every 30 min by using TLC (solvent system—chloroform: methanol; 1:1). After 12 h and the confirmation of the completion of the reaction by TLC, the reaction mixture was then cooled, and cold water (30 mL) was added. Filtration of the product followed by washing with cold water frequently 4 to 5 times and further recrystallization in ethanol provided compound [22,23,24,25,26].

White crystalline and solid; yield% = 95%; m.p. = 210 °C; solubility; insoluble: water, acetic acid glacial; partially soluble: ethanol and methanol; fully soluble: DMSO, chloroform. DMSO and methanol, chloroform. IRv_max_ (cm^−1^; KBr pellets); 1770.71, 1701.27 (C=O imide); 1336.71, 1151.54 (S=O) and 3363.97 (NH_2_), 2987.84 (aliphatic CH_2_). Mass (ESI-); m/z: 278.6341 [M−H]. ^1^H NMR (400 MHz, DMSO-d_6_): δ 7.025 (s, 2H, SO_2_NH_2_), 7.335 (d, 2H, Ar−H from pyrrol), 7.386−7.407, 7.745−7.765 (d, 4H, Ar−H from nitrobenzene), 2.922−2.958, 3.695−3.731 (t, 2H, CH_2_ aliphatic). ^13^C NMR (100 MHz, DMSO−d_6_): 34.33, 38.99, 126.63, 130.06, 135.38, 143.26, 143.28, 171.66. Elem. anal. calculated for C_12_H_12_N_2_O_4_S: C, 51.42; H, 4.32; N, 9.99; O, 22.83; S, 11.44; found C, 51.41; H, 4.317; N, 9.99; O, 22.83; S, 11.34.

#### 3.2.3. Synthesis of 2-(4,5,6,7-Tetrachloro-1,3-dioxoisoindolin-2-yl) Benzo[d]Thiazole-5-Sulphonamide (**3**)

A reaction containing 0.002 moles (0.558 g) of 2-aminobenzo[d]thiazole-5-sulphonamide and 0.002 moles (0.572 g) of 4,5,6,7-tetrachlorophthalic anhydride in glacial acetic acid (as solvent) was stirred at 130^°^C for 12 h under nitrogen environment to produce 0.002 moles of 2-(4,5,6,7-tetrachloro-1,3-dioxoisoindolin-2-yl) benzo[d]thiazole-5-sulphonamide (**3**) (0.989 g). The reaction was checked every 30 min by using TLC (solvent system—chloroform: methanol; 1:1). After 12 h and the confirmation of the completion of the reaction by TLC, the reaction mixture was then cooled, and cold water (30 mL) was added. Filtration of the product followed by washing with cold water frequently 4 to 5 times and further recrystallization in ethanol provided compound [22,23,24,25,26].

White amorphous and solid; yield% = 87.96%; m.p. = 302.2 °C; solubility; insoluble: acetic acid glacial; partially soluble: ethanol, methanol; fully soluble: water, DMSO and chloroform. IRv_max_ (cm^−1^; KBr pellets); 1730.21, 1776.50 (C=O imide); 1159.26, 1340.57 (S=O) and 3265.59 (NH_2_). Mass (ESI+); m/z: 498.3121 [M+1]. ^1^H NMR (400 MHz, DMSO−d_6_): δ 7.537 (s, 2H, SO_2_NH_2_), 8.022−8.048, 8.239−8.261 (d, 2H, Ar−H from benzothiazole sulphonamide), 8.754−8.758 (s, H, Ar−H from benzothiazole sulphonamide). ^13^C NMR (100 MHz, DMSO−d_6_): 121.51, 123.81, 125.21, 128.8, 129.73, 129.91, 133.46, 134.11, 134.71, 140.2, 141.88, 151.62, 155.45.

#### 3.2.4. Synthesis of 2-(4,5,6,7-Tetrabromo-1,3-dioxoisoindolin-2-yl) Benzo[d]Thiazole-5-Sulphonamide (4)

A reaction containing 0.002 moles (0.558 g) of 2-aminobenzo[d]thiazole-5-sulphonamide and 0.002 moles (0.919 g) of 4,5,6,7-tetrabromophthalic anhydride in glacial acetic acid (as solvent) was stirred at 130 °C for 12 h under nitrogen environment to produce 0.002 moles of 2-(4,5,6,7-tetrachloro-1,3-dioxoisoindolin-2-yl) benzo[d]thiazole-5-sulphonamide (**4**) (1.341 g). The reaction was checked every 30 min by using TLC (solvent system—chloroform: methanol; 1:1). After 12 h and the confirmation of the completion of the reaction by TLC, the reaction mixture was then cooled, and cold water (30 mL) was added. Filtration of product followed by washing with cold water frequently 4 to 5 times and further recrystallization in ethanol provided compound [22,23,24,25,26]

Reddish white amorphous and solid; yield% = 51%; m.p. = 291.6 °C; solubility; insoluble: water, acetic acid glacial; partially soluble: ethanol and methanol; fully soluble: DMSO, chloroform. IRv_max_ (cm^−1^; KBr pellets); 1739.85, 1786.14 (C=O imide); 1338.64, 1163.11 (S=O) and 3306.10 (NH_2_). Mass (ESI+); m/z: 674.6411 [M+1]. ^1^H NMR (400 MHz, DMSO−d_6_): δ 7.531 (s, 2H, SO_2_NH_2_), 8.018−8.044, 8.225−8.246 (d, 2H, Ar−H from benzothiazole sulphonamide), 8.745−8.749 (s, H, Ar−H from benzothiazole sulphonamide).^13^C NMR (100 MHz, DMSO−d_6_): 121.48, 122.41, 123.74, 125.17, 131.54, 133.49, 136.98, 138.55, 141.8, 151.67, 155.66, 161.29, 166.76.

#### 3.2.5. Synthesis of 3-Fluoro-4-(5-Nitro-1,3-dioxoisoindolin-2-yl) Benzenesulphonamide (5)

A reaction containing 0.002 moles (0.380 g) of 4-amino-2-fluorobenzenesulphonamide and 0.002 moles (0.386 g) of 5-nitroisobenzofuran-1,3-dione in glacial acetic acid (as solvent) was stirred at 130 °C for 12 h under nitrogen environment to produce 0.002 moles of 3-fluoro-4-(5-nitro-1,3-dioxoisoindolin-2-yl) benzenesulphonamide (**5**) (0.720 g). The reaction was checked every 30 min by using TLC (solvent system—chloroform: methanol; 1:1). After 12 h and the confirmation of the completion of the reaction by TLC, the reaction mixture was then cooled, and cold water (30 mL) was added. Filtration of the product followed by washing with cold water frequently 4 to 5 times and further recrystallization in ethanol provided compound [22,23,24,25,26].

White crystalline and solid; yield% = 85%; m.p. = 232 °C; solubility; insoluble: water, acetic acid glacial; partially soluble: ethanol; fully soluble: DMSO, chloroform and methanol. IRv_max_ (cm^−1^; KBr pellets); 1788.70, 1733.10 (C=O imide); 1349.25, 1162.15 (S=O) and 3369.75 (NH_2_), 1541.18, 1386.86 (NO_2_). Mass (ESI-); m/z: 364.1251 [M−H]^−^. ^1^H NMR (400 MHz, DMSO−d_6_): δ 7.689 (s, 2H, SO_2_NH_2_), 7.834−7.872, 7.895−7.916 (d, 2H, Ar−H from benzenesulphonamide), 7.936–7.94 (s, H, Ar−H from benzenesulphonamide), 8.314−8.335, 8.699−8.773 (d, 2H, Ar−H from nitrobenzene), 8.694−8.695 (s, 2H, Ar−H from nitrobenzene). ^19^F NMR (300 MHz, DMSO−d_6_): δ -116.373.^13^C NMR (100 MHz, DMSO−d_6_): 114.95, 115.17, 119.61, 122.82, 122.95, 123.26, 123.30, 126.31, 131.08, 132.15, 133.81, 137.01, 147.40, 147.46, 152.69, 156.30, 158.84, 164.96, 165.21.

#### 3.2.6. Synthesis of 4-(4,5,6,7-Tetrachloro-1,3-dioxoisoindolin-2-yl) Benzenesulphonamide (6)

An amount of 0.344 g (0.002 moles) 4-amino benzenesulphonamide stirred under nitrogen environment with 0.572 g (0.002 moles) of 4,5,6,7-tetrachlorophthalic anhydride to produce 0.880 g (0.002 moles) of 4-(4,5,6,7-tetrachloro-1,3-dioxoisoindolin-2-yl) benzenesulphonamide (**6**) in the presence of glacial acetic acid as solvent for 2 h at 130 °C. The reaction was monitored each 30 min with the help of TLC (chloroform: methanol; 3:1). After that the mixture was cooled and 30 mL of cold water was added. The product was filtered and washed with cold water repeatedly for 4/5 times. The product was recrystallised in ethanol to purify the synthesised compound **6**.

White crystalline and solid; yield% = 98%; mp = 315–320 °C (decompose); solubility; insoluble: water, acetic acid glacial; partially soluble: ethanol; fully soluble: DMSO and methanol, chloroform. IRv_max_ (cm^−1^; KBr pellets); 1778.43, 1716.70 (C=O imide); 1321.28, 1163.11 (S=O) and 3414.12 (NH_2_). Mass (ESI+); m/z: 438.8015 [M+H]^+^. ^1^H NMR (400 MHz, DMSO-d6): δ_H_ 7.533 (s, 2H, SO_2_NH_2_), 7.699–7.678 (d, 2H, Ar-H from benzenesulphonamide), 8.050–8.029 (d, 2H, Ar-H from benzenesulphonamide). ^13^C NMR (100 MHz, DMSO-d6): δ_C_ 167.29, 144.38, 138.98, 134.49, 128.9, 128.8, 128.23, 127.04.

#### 3.2.7. Synthesis of 4-((4,5,6,7-Tetrachloro-1,3-dioxoisoindolin-2-yl) Methyl) Benzenesulphonamide (7)

An amount of 0.445 g (0.002 moles) of 4-(aminomethyl)benzenesulphonamide hydrochloride stirred under nitrogen environment with 0.572 g (0.002 moles) of 4,5,6,7-tetrachlorophthalic anhydride to produce 0.980 g (0.002 moles) of 4-(2-(4,5,6,7-tetrachloro-1,3-dioxoisoindolin-2-yl) ethyl)benzenesulphonamide hydrochloride (**7**) in the presence of glacial acetic acid as solvent for 1.5 h at 130 °C. The reaction was monitored each 30 min with the help of TLC (chloroform: methanol; 1:1). After that the mixture was cooled and 30 mL of cold water was added. The product was filtered and washed with cold water repeatedly for 4/5 times. The product was recrystallised in ethanol to purify the synthesised compound **7**.

White crystalline and solid; yield% = 90%; mp = 305 °C; Solubility; insoluble: water, acetic acid glacial; partially soluble: ethanol and methanol; fully soluble: DMSO, chloroform. IRv_max_ (cm^−1^; KBr pellets); 1774.57, 1714.77 (C=O imide); 1338.64, 1161.19 (S=O) and 3350.46 (NH_2_). Mass (ESI+); m/z: 451.8231 [M], 452.8101 [M+H]^+^. ^1^H NMR (400 MHz, DMSO-d6): δ_H_ 4.891 (s, 2H, CH_2_), 7.38 (s, 2H, SO_2_NH_2_), 7.565, 7.586 (d, 2H, Ar-H from benzenesulphonamide), 7.81, 7.831 (d, 2H, Ar-H from benzenesulphonamide). ^13^C NMR (100 MHz, DMSO-d6): δ_C_ 164.28, 144.26, 140.63, 139.1, 129.49, 129.1, 128.89, 126.81, 42.1.

#### 3.2.8. Synthesis of 4-(2-(4,5,6,7-Tetrachloro-1,3-dioxoisoindolin-2-yl) Ethyl) Benzenesulphonamide (8)

An amount of 0.400 g (0.002 moles) of 4-(2-aminoethyl) benzenesulphonamide stirred under nitrogen environment with 0.572 g (0.002 moles) of 4,5,6,7-tetrachlorophthalic anhydride to produce 0.936 g (0.002 moles) of 4-(2-(4,5,6,7-tetrachloro-1,3-dioxoisoindolin-2-yl)ethyl)benzenesulphonamide (**8**) in the presence of glacial acetic acid as solvent for 1.5 h at 130 °C. The reaction was monitored each 30 min with the help of TLC (chloroform: methanol; 1:1). After that the mixture was cooled and 30 mL of cold water was added. The product was filtered and washed with cold water repeatedly for 4/5 times. The product was recrystallised in ethanol to purify the synthesised compound **8**.

White crystalline and solid; yield% = 98%; mp = 286 °C; solubility; insoluble: water, acetic acid glacial; partially soluble: ethanol, methanol; fully soluble: DMSO and chloroform. IRv_max_ (cm^−1^; KBr pellets); 1776.50, 1712.92 (C=O imide); 1303.92, 1159.29 (S=O) and 3387.11 (NH_2_), 2955.04 (aliphatic CH_2_). Mass (ESI+); m/z: 465.2237 [M]^+^. ^1^H NMR (400 MHz, DMSO-d6): δ_H_ 3.011–3.048 (t, 2H, CH_2_), 3.849–3.885 (t, 2H, CH_2_), 7.347 (s, 2H, SO_2_NH_2_), 7.473–7.494 (d, 2H, Ar-H from benzenesulphonamide), 7.763–7.785 (d, 2H, Ar-H from benzenesulphonamide). ^13^C NMR (100 MHz, DMSO-d6): δ_C_ 164.19, 143.41, 138.143.21, 139.17, 130.16, 129.23, 129.03, 126.81, 34.13.

#### 3.2.9. Synthesis of 3-(4,5,6,7-Tetrachloro-1,3-dioxoisoindolin-2-yl) Benzenesulphonamide (9)

An amount of 0.344 g (0.002 moles) of 3-amino benzenesulphonamide stirred under nitrogen environment with 0.572 g (0.002 moles) of 4,5,6,7-tetrachlorophthalic anhydride to produce 0.880 g (0.002 moles) of 3-(4,5,6,7-tetrachloro-1,3-dioxoisoindolin-2-yl) benzenesulphonamide (**9**) in the presence of glacial acetic acid as solvent for 2 h at 130 °C. The reaction was monitored each 30 min with the help of TLC (chloroform: methanol; 3:1). After that the mixture was cooled and 30 mL of cold water was added. The product was filtered and washed with cold water repeatedly for 4/5 times. The product was recrystallised in ethanol to purify the synthesised compound **9**.

White crystalline and solid; yield% = 92%; mp = 308 °C; Solubility; insoluble: water, acetic acid glacial; partially soluble: ethanol and methanol; fully soluble: DMSO, chloroform. IRv_max_ (cm^−1^; KBr pellets); 1782.29, 1718.68 (C=O imide); 1301.99, 1132.25 (S=O) and 3356.25 (NH_2_). Mass (ESI+); m/z: 437.7912 [M], 438.7103 [M+H]^+^. ^1^H NMR (400 MHz, DMSO-d6): δ_H_ 7.61 (s, 2H, SO_2_NH_2_), 7.7–7.72 (s, H, Ar-H from benzenesulphonamide), 7.799–7.84 (t, H, Ar-H from benzenesulphonamide), 7.969–7.988 (d, 2H, Ar-H from benzenesulphonamide). ^13^C NMR (100 MHz, DMSO-d6): δ_C_ 163.42, 145.99, 139.4, 132.56, 131.68, 130.92, 129.39, 126.86, 125.6.

#### 3.2.10. Synthesis of 2-(4,5,6,7-Tetrachloro-1,3-dioxoisoindolin-2-yl) Benzenesulphonamide (10)

An amount of 0.344 g (0.002 moles) of 2-amino benzenesulphonamide stirred under nitrogen environment with 0.572 g (0.002 moles) of 4,5,6,7-tetrachlorophthalic anhydride to produce 0.880 g (0.002 moles) of 2-(4,5,6,7-tetrachloro-1,3-dioxoisoindolin-2-yl) benzenesulphonamide (**10**) in the presence of glacial acetic acid as solvent for 2 h at 130 °C. The reaction was monitored each 30 min with the help of TLC (chloroform: methanol; 3:1). After that the mixture was cooled and 30 mL of cold water was added. The product was filtered and washed with cold water repeatedly for 4/5 times. The product was recrystallised in ethanol to purify the synthesised compound **10**.

White crystalline and solid; yield% = 90%; mp = 313 °C; Solubility; insoluble: water, acetic acid glacial; partially soluble: ethanol and methanol; fully soluble: DMSO and chloroform. IRv_max_ (cm^−1^; KBr pellets); 1782.29, 1712.85 (C=O imide); 1305.85, 1139.97 (S=O) and 3387.11 (NH_2_). Mass (ESI+); m/z: 437.7154 [M]^+^, 438.7152 [M+H]^+^. ^1^H NMR (400 MHz, DMSO-d6): δ_H_ 7.661 (s, 2H, SO_2_NH_2_), 7.802–7.844 (t, 2H, Ar-H from benzenesulphonamide), 8.102–8.107, 8.1–22-8.126 (d, 2H, Ar-H from benzenesulphonamide). ^13^C NMR (100 MHz, DMSO-d6): δ_C_ 165.71, 163.11, 143.27, 139.53, 134.05, 132.74, 131.63, 129.62, 129.48, 129.26.

#### 3.2.11. Synthesis of 3-Fluoro-4-(4,5,6,7-Tetrachloro-1,3-dioxoisoindolin-2-yl) Benzenesulphonamide (11)

An amount of 0.380 g (0.002 moles) of 4-amino-2-fluoro benzenesulphonamide stirred under nitrogen environment with 0.572 g (0.002 moles) of 4,5,6,7-tetrachlorophthalic anhydride to produce 0.916 g (0.002 moles) of 3-fluoro-4-(4,5,6,7-tetrachloro-1,3-dioxoisoindolin-2-yl) benzenesulphonamide (**11**) in the presence of glacial acetic acid as solvent for 12 h at 130 °C. The reaction was monitored each 30 min with the help of TLC (chloroform: methanol; 3:1). After that the mixture was cooled and 30 mL of cold water was added. The product was filtered and washed with cold water repeatedly for 4/5 times. The product was recrystallised in ethanol to purify the synthesised compound **11**.

White crystalline and solid; yield% = 95%; mp = 278 °C; Solubility; insoluble: water, acetic acid glacial; partially soluble: ethanol; fully soluble: DMSO, chloroform and methanol. IRv_max_ (cm^−1^; KBr pellets); 1790.00, 1720.56 (C=O imide); 1301.99, 1168.90 (S=O) and 3385.18 (NH_2_). Mass (ESI+); m/z: 455.7312 [M]^+^, 456.7317 [M+H]^+^. ^1^H NMR (400 MHz, DMSO-d6): δ_H_ 7.704 (s, 2H, SO_2_NH_2_), 7.796–7.835 (m, H, Ar-H from benzenesulphonamide), 7.893–7.941 (d, 2H, Ar-H from benzenesulphonamide). ^13^C NMR (100 MHz, DMSO-d6): δ_C_ 162.74, 159.07, 156.53, 147.76, 139.89, 132.35, 129.78, 123.52, 122.74, 115.39.

#### 3.2.12. Synthesis of 3-Chloro-4-(4,5,6,7-Tetrachloro-1,3-Dioxoisoindolin-2-yl) Benzenesulphonamide (12)

An amount of 0.412 g (0.002 moles) of 4-amino-2-chloro benzenesulphonamide stirred under nitrogen environment with 0.572 g (0.002 moles) of 4,5,6,7-tetrachlorophthalic anhydride to produce 0.949 g (0.002 moles) of 3-chloro-4-(4,5,6,7-tetrachloro-1,3-dioxoisoindolin-2-yl)benzenesulphonamide (**12**) in the presence of glacial acetic acid as solvent for 12 h at 130 °C. The reaction was monitored each 30 min with the help of TLC (chloroform: methanol; 3:1). After that the mixture was cooled and 30 mL of cold water was added. The product was filtered and washed with cold water repeatedly for 4/5 times. The product was recrystallised in ethanol to purify the synthesised compound **12**.

White crystalline and solid; yield% = 94%; mp = 303 °C; solubility; insoluble: water, acetic acid glacial; partially soluble: ethanol; fully soluble: DMSO, chloroform and methanol. IRv_max_ (cm^−1^; KBr pellets); 1788.07, 1722.49 (C=O imide); 1305.87, 1172.76 (S=O) and 3387.11 (NH_2_). Mass (ESI+); m/z: 472.6115 [M+H]^+^. ^1^H NMR (400 MHz, DMSO-d6): δ_H_ 7.724 (s, 2H, SO_2_NH_2_), 7.861–7.882 (d, H, Ar-H from benzenesulphonamide), 8.017–8.042 (m, H, Ar-H from benzenesulphonamide), 8.155–8.16 (d, H, Ar-H from benzenesulphonamide). ^13^C NMR (100 MHz, DMSO-d6): δ_C_ 162.64, 147.69, 140.16, 133.91, 133.12, 132.82, 129.91, 129.11, 129.31, 126.65.

#### 3.2.13. Synthesis of 3-Bromo-4-(4,5,6,7-Tetrachloro-1,3-Dioxoisoindolin-2-yl) Benzenesulphonamide (13)

An amount of 0.502 g (0.002 moles) of 4-amino-2-bromo benzenesulphonamide stirred under nitrogen environment with 0.572 g (0.002 moles) of 4,5,6,7-tetrachlorophthalic anhydride to produce 1.038g (0.002 moles) of 3-bromo-4-(4,5,6,7-tetrachloro-1,3-dioxoisoindolin-2-yl) benzenesulphonamide (**13**) in the presence of glacial acetic acid as solvent for 12 h at 130 °C. The reaction was monitored each 30 min with the help of TLC (chloroform: methanol; 3:1). After that the mixture was cooled and 30 mL of cold water was added. The product was filtered and washed with cold water repeatedly for 4/5 times. The product was recrystallised in ethanol to purify the synthesised compound **13**.

White crystalline and solid; yield% = 90%; mp = 265 °C; Solubility; insoluble: Water, acetic acid glacial; partially soluble: ethanol; fully soluble: DMSO, chloroform and methanol. IRv_max_ (cm^−1^; KBr pellets); 1772.64, 1720.56 (C=O imide); 1300.00, 1170.83 (S=O) and 3400.0 (NH_2_). Mass (ESI+); m/z: 516.5214 [M+H]^+^. ^1^H NMR (400 MHz, DMSO-d6): δ_H_ 7.715 (s, 2H, SO_2_NH_2_), 7.842–7.863 (d, H, Ar-H from benzenesulphonamide), 8.051–8.077 (d, H, Ar-H from benzenesulphonamide), 8.292–8.297 (d, H, Ar-H from benzenesulphonamide). ^13^C NMR (100 MHz, DMSO-d6): δ_C_ 162.64, 147.69, 140.16, 133.91, 133.12, 132.82, 129.91, 129.11, 129.31, 126.65.

### 3.3. CA Inhibition Assay

To check the CA-catalyzed CO_2_ hydration activity an applied photo physics stopped-flow instrument has been used. A phenol red indicator (0.2 mM) has been used at the λ_max_ of 557 nm, with 20 mM Hepes buffer (pH 7.5) and Na_2_SO_4_ (20 mM), following the initial rates of the CA-catalysed CO_2_ hydration reaction for 10 to100 s. To determine the kinetic parameters and inhibition constants, 1.7 to 17 mM CO_2_ concentrations were used. To determine each inhibitor’s initial velocity minimum of six traces have been used from the initial 5–10% of the reaction. A similar process has been applied to determine the uncatalyzed rates. Further, the uncatalyzed rates were deducted from the total observed rates found. 0.1 mM solution of stock inhibitors were prepared, and dilutions of up to 0.01 nM were done. A preincubation of inhibitor and CA solutions was done for 15 min at room temperature for allowing the formation of the enzyme-inhibitor adduct. The PRISM 3 software was used to obtain inhibition constants by nonlinear least-squares methods [30].

### 3.4. Molecular Docking Study

Molecular docking experiment was executed in Maestro Glide of Schrödinger [39,40]. GLIDE XP was used for the study of protein-ligand interactions of the hCA isozymes. The best pose of each ligand was ranked, and the docking score is based on Chem Score. The co-crystallised ligand with RMSD value of 1.5 Å, representing that GLIDE is an impeccable platform for docking study of hCAIs.

## 4. Conclusions

The synthesis and characterization of a series of sulphonamides **1–13** are described. The reaction between amino group and anhydrides involves S_N_^2^ type reaction mechanism, in which amino-containing sulphonamide derivatives react with aliphatic/aromatic anhydrides. These sulphonamides were tested against four physiologically similar hCA I, II, IX, and XII isoforms. The relationships between the chemical structure of a molecule and its hCA inhibition activity (SAR) were established and rationalised with the help of molecular docking studies. These sulphonamides were potent inhibitors of hCA I and II in nM inhibition constant. The compounds inhibited the transmembrane isoforms hCA IX and XII with moderate to high nM concentration. The sulphonamides reported here had a moderate to poor selectivity ratio. Therefore, these are better hCA I and II selective inhibitors than hCA IX and XII. Such compounds may be useful as an interesting candidate for the drug development of highly selective and efficacious novel sulphonamides for hypoxia-induced cancer drug therapy and retinopathies. The sulphonamides which potentially inhibited hCA I and II can be further developed as a novel interesting clinical candidate for the treatment of retinal/cerebral oedema, ischemic diabetic cardiomyopathy, coronary revascularization, diabetic retinopathy, epilepsy, cancer, altitude sickness, COVID-19, etc.

## Data Availability

Data available within the article.

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
