# Peer review of "Synthesis and Human Carbonic Anhydrase I, II, IX, and XII Inhibition Studies of Sulphonamides Incorporating Mono-, Bi- and Tricyclic Imide Moieties"

_pharmaceuticals, 2021, doi:10.3390/ph14070693_

Round 1
Reviewer 1 Report
Dear Authors,
after a careful reading i would like to present some suggestion to be made in the data presentation:
- There is statistically significant data suggested in table without the error of mean while presented the explanation under the table with steric, I would suggest to include the statistical analysis of the Ki values of inhibitors.
- The table for Ki values of the molecules can be separated from their docking data.
After these little modifications, the article can be re submit to proceed further.
Best regards,
Saima
Reviewer 2 Report
Authors demonstrated the the effect of anti-human 15 carbonic anhydrases (hCAs, EC 4.2.1.1) I, II, IX and XII of novel synthesized sulphonamides by reaction of amino-containing aromatic sulphon- 14 amides with mono-, bi- and tricyclic anhydrides. The study is well designed, well presented.
This study can be published after minor revision.
- The letters and numbers in figure 4 should be enlarged. Also, "a." "b." signs in the figure 4 should be moved.
- "3.2.12. Synthesis of 3-bromo-4-(4,5,6,7-tetrachloro-1,3- ioxoisoindolin-2-yl) benzenesulphonamide (13) " in line 526 should be "3.2.13. Synthesis of 3-bromo-4-(4,5,6,7-tetrachloro-1,3- ioxoisoindolin-2-yl) benzenesulphonamide (13) "
- Please describe more about the different drug designs including ligand-based, pharmacophore based and structure based designs. What are their pros and cons?
Author Response
Please see the attachment

This manuscript is a resubmission of an earlier submission. The following is a list of the peer review reports and author responses from that submission.